# The Sexual Economies of Clericalism: Women Religious and Gendered Violence in the Catholic Church

**Kathleen McPhillips *** and **Tracy McEwan ***

School of Humanities, Creative Industries and Social Science, University of Newcastle,
Callaghan, NSW 2308, Australia
* Correspondence: kathleen.mcphillips@newcastle.edu.au (K.M.); tracy.mcewan@newcastle.edu.au (T.M.)

**Abstract:** As the sexual abuse crisis continues to plague the Catholic Church across the world, the focus on men as both perpetrators of sexual violence, and victims of child sexual abuse has been at the forefront of media and academic analysis. However, evidence from public inquiries, criminal investigations, academic research and survivor testaments identifies women religious as both perpetrators of sexual and physical violence against children and victims of clerical sexual violence. This indicates that women religious, colloquially known as nuns, belong to a very unsettled landscape in religious gender politics in that they have been both victims of male clerical abuse and perpetrators of child abuse. This article considers these two contested realities by examining and analysing the evidence from research studies and public inquiries. We found that nuns were not only perpetrators of physical and sexual violence against children, but they were also engaged in herding children into the pathways of organised clerical paedophiles, particularly in children's homes and orphanages where children were considered sexual commodities. Simultaneously, the sexual abuse of nuns has recently become a major concern for religious communities and the wider Church. We argue that by bringing these two realities together through a new discourse—the sexual economies of clericalism—new understandings of the agency of nuns can be explored in a wider theoretical and methodological framework.

**Keywords:** nuns; women religious; sexual abuse; clericalism; gendered violence; Catholic Church

(**Content Warning:** This article discusses examples and cases of child sexual abuse and the sexual abuse of women religious. Readers take note before reading.)

## 1. Introduction

The role of Catholic nuns[1] in relation to the sexual abuse crisis has been muted, even hidden, throughout the numerous world-wide public and Church-based enquiries of the last 20 years. More recently, accounts of nuns as firstly victims of widespread sexual abuse by Catholic clerics and secondly in their role in facilitating child abuse across Catholic institutions has come to light through the work of public inquiries, activist groups, media reports and research.[2]

Research in this field provides evidence of two central facts. Firstly, women religious have been subject to significant levels of widespread sexual abuse by clerical perpetrators across the globe for much of the 20th century. The extent of this abuse has been known for decades yet remains poorly recognized and addressed (Angulo 2021; Brock 2019; Chibnall et al. 1998; Cucci 2020; Horowitz 2019; Marcel 2013; Tuttle 2020). Secondly, women religious were instrumental in facilitating the networks of clerical perpetrators particularly in providing access to children, as well as abusing children physically and sexually (Donovan 2002; The Inquiry into Historical Institutional Abuse 1922 to 1995 and The Executive Office (HIAI) 2017; Hidalgo 2007; Royal Commission into Institutional Responses to Child Sexual Abuse (RCIRCSA) 2016, 2017c, 2017d; Scottish Child Abuse Inquiry (SCAI) 2018, 2019; Te Rōpū Tautoko 2022).

In this article we will argue that women religious carry the double burden of being both infantilized and at risk of abuse in a patriarchal male celibate Catholic culture yet have also facilitated and instigated child abuse. While current analysis tends to concentrate on one of these thematics, we will attempt to bring both these contexts together to produce a comprehensive understanding of the culture and reality of the conditions in which women religious have lived and worked in late modernity. Our central research questions are firstly what are the socio-theological conditions in which the agency of women religious is formed? And secondly what are the specific forms of gendered violence that are operationalised in these conditions?

Methodologically, we provide a substantive review of the literature in the field of gendered violence and women religious looking at both investigative studies and methodological approaches. We then draw on this research to propose a new theoretical framework—the sexual economies of clericalism—to analyse the social and gendered conditions that support cultures of sexual abuse. This framework locates clericalism,[3] abuse and its economies in the circulation of patriarchal power that typifies Catholic institutional culture, which we name as gendered violence. Within these economies women religious have forms of agency that are mediated by the particular cultural, spiritual and political conditions in which they live and work. We will draw on analysis covering the modern histories and roles of women religious and female religious orders and institutes in relation to gendered patterns of clericalist behaviour and theology. We examine the position and agency of women religious in a clericalist masculinist Church and attempt to address injustices and harm perpetrated against women and children. Clearly, the contradictions that such a system of gendered violence produces where nuns are both facilitators and perpetrators or child sexual abuse (CSA), as well as victims of sexual abuse require careful investigation and attention.

The article will begin with a literature review covering the sexual abuse of nuns and the evidence of their role in facilitating the abuse of children. Following this, the concept of the sexual economies of clericalism will be considered and then applied to the forms of gendered violence described above.

## 2. The Sexual Abuse of Women Religious: Bringing a Travesty to Light

The sexual abuse of women in religious life has not been subject to the same levels of scrutiny as that given to the sexual abuse of children. Despite several important investigations dating from the mid-1990s which clearly established the abuse of nuns as a serious issue for the Church and provided recommendations addressing remediation (Allen and Schaeffer 2001; Chibnall et al. 1998; M. McDonald 1998; O'Donohue 1994, 1995), none of the recommendations were acted upon and further research was not followed up. In her report, O'Donohue (1994) documented widespread sexual abuse of religious women by male clerics across the globe, and provided multiple recommendations to address the issue. A few years later, M. McDonald (1998) reported the sexual and spiritual abuse of religious sisters in Africa and called for "concerted action" to break a "conspiracy of silence". Yet, even though the O'Donohue (1994) report was authorized by the Vatican and followed up by a memo which made clear recommendations (O'Donohue 1995), and M. McDonald (1998) directly addressed various Vatican committees, concerns were not acted on, leaving the scandal of the sexual abuse of women in religious life to remain concealed and obscured by Catholic Church leaders (Figueroa and Tombs 2022; Hidalgo 2007; Horowitz 2019; Wagner 2018). In 2001, the inaction by the Vatican became public when the National Catholic Reporter released both reports (Allen and Schaeffer 2001). This led to numerous media reports and a response by the Vatican indicating that it was aware of the problem and was addressing it (Wagner 2018). However, the issue was eclipsed by the Boston Globe revelation of CSA and subsequently shelved until it was raised again in 2018 (Figueroa and Tombs 2022). Since 2018, a new awareness of this travesty has been visible as victims/survivors break their silence and verbalise their experiences in public events (further research documented below).

At a 2018 Voices of Faith event German theologian and former nun, Doris Wagner spoke out about being raped by a priest who was the superior of her convent (Figueroa and Tombs 2022; Voices of Faith 2018b). The February 2019 issue of "Women Church World", which is distributed as part of the Vatican newspaper L'Osservatore Romano, included an article on the sexual abuse of women religious by priests (Gamble 2019). Editor Lucetta Scaraffia wrote:

> If the church continues to close its eyes to the scandal—made even worse by the fact that abuse of women brings about procreation and is therefore at the origin of forced abortions and children who aren't recognised by priests—the condition of oppression of women in the church will never change. (cited in Gamble 2019)

In February 2019, during a press conference, the current pope, Francis, acknowledged that priests and bishops have been responsible for the sexual manipulation of women religious (Francis 2019).

In 2022, a well-publicised court case against Bishop Franco Mulakkal in Kerala, India heard evidence that he had raped a nun multiple times in her convent during 2014–2016. The court case ruled in his favour but raised significant concerns regarding the safety of Catholic women religious in India and their capacity to disclose to civil authorities (Ellis-Petersen 2022). This new visibility of gendered violence in the Catholic Church represents an important opportunity to advance evidence-based research and knowledge on this topic via sociological and theological analysis.

### 2.1. Historical Evidence

There is some historical evidence of women religious being subjected to various forms of physical, psychological and sexual abuse perpetrated by clerical men in the Catholic Church during the 19th and 20th century. This evidence is largely in narrative form and lacks corroborating evidence. For example, Mary MacKillop (1842–1909) Australia's first saint and co-founder of Australia's first order of women religious—the Institute of the Sisters of Saint Joseph of the Sacred Heart—is recognised for her dedication to social justice issues, including gender equality in education, and the care of young women (Lewis 2010; McPhillips 2013). However, her steadfast commitment to the self-governance of her order created conflict with the local clerical authorities in Adelaide. In 1871 she was excommunicated by the local bishop on the grounds she was an alcoholic. It was later revealed that MacKillop's excommunication was, in large part, because she raised the alarm of the sexual abuse of children by a local priest and of the abuse of women in her order by priests. In September 1871, around the time of MacKillop's excommunication, Sister Mary Angela (formally Catherine Carroll) wrote to MacKillop disclosing her sexual and spiritual abuse by a senior priest. Mary Angela writes that a priest, Father Horan:

> took my hand in his and drew me to him and tried to put his arms round my waist and told me not to be afraid of him . . . he would not let me go away from him till I confessed it, and if I would not any hour might find me in the deepest pit in hell. (cited in Condon 2000, p. 1816)

Later in the letter, Mary Angela recalls how Father Horan entered her bedroom and accosted her. In her letter of 1871 to Mary Mackillop, Sister Mary Angela describes feeling ill and frightened yet would be eventually discredited and labelled "insane" (Condon 2000).

### 2.2. Knowledge of Current Sexual Abuse

There has been limited academic research into the sexual abuse of religious women. The most broad-ranging English-language studies are the research projects conducted by Chibnall et al. (1998), D'Lima et al. (2021), Figueroa and Tombs (2020, 2021, 2022) and Durà-Vilà et al. (2013).

The national survey of US Catholic nuns detailed in Chibnall et al. (1998) estimated that three in ten (30 per cent) nuns surveyed experienced sexual trauma in religious life, with more than one in ten (11.1 per cent) experiencing sexual exploitation or coercion.

It established 'the most common type of exploiter was a priest . . . acting in the role of spiritual director' (Chibnall et al. 1998, p. 151). Of the nuns who described their sexual exploitation in the survey the vast majority (89.7 per cent) experienced physical sexual contact, for example touching, physical closeness, kissing, fondling, oral sex, or sexual intercourse (Chibnall et al. 1998). For slightly fewer than a third (29.5 per cent) the sexual exploitation was a one-off event, however for just less than a half (46.6 per cent) it lasted between 1 month and 2 years, and for around a fifth (19.2 per cent) it lasted more than 3 years (Chibnall et al. 1998). Chibnall and colleagues state that:

> The results of the study suggest that Catholic sisters in the United States are no strangers to sexual trauma. With a lifetime prevalence of 40% and a prevalence during religious life of nearly 30%, sexual trauma in one form or another impacted a minimum of 34,000 of the 85,000 Sisters who were in active at the time of the study (projected), with sometimes significant psychological and spiritual consequences. (Chibnall et al. 1998, p. 158)

Sexual harassment also constitutes an unsafe working environment for many Catholic nuns. In their study, Chibnall et al. (1998) found that slightly less than one in ten (9.3 per cent) of nuns surveyed claimed to be the focus of workplace sexual harassment at least once since joining religious life; with the harassment primarily perpetrated by priests or other men and women religious. Cases tended to involve some type of physical contact, with genital contact significantly more likely if the harasser was a priest or nun rather than a lay person (Chibnall et al. 1998).

In 2020, women religious in India published their experiences of gendered based spiritual, emotional and sexual harm via a study commissioned by the Conference of Religious India (CRI) (D'Lima et al. 2021). The study used a questionnaire to survey 121 women religious. The qualitative and quantitative data gathered revealed a high level of harassment of women religious in India, including verbal and sexual abuse (D'Lima et al. 2021).

Four research projects between 2016–2020 by Figueroa and Tombs (2020, 2021, 2022) involved interviews with male and female consecrated survivors in South America, Europe and the Philippines and reported on experiences of sexual and spiritual abuse utilising a particular theological framework reading the crucifixion of Jesus as sexual abuse with the aim of developing a sensitive pastoral response (Tombs 2019).

Durà-Vilà et al. (2013) explores the experiences and coping strategies of five Spanish women religious who are victim/survivors of sexual abuse by priests. In the ethnographic study, participants described various forms of sexual trauma, 'involving violence, exploitation, coercion, and mental and physical stress' (Durà-Vilà et al. 2013, p. 25). All the women religious suffered from an 'intense negative emotional response following the abuse, experiencing the sexual advances as something "terrible" and "unthinkable", completely outside the usual range of human experience' (Durà-Vilà et al. 2013, p. 26). Durà-Vilà et al. (2013) report that abuse shattered implicit beliefs regarding the integrity of priests and their safety in the Church.

There are a further four recent and notable studies that are yet to be translated into English, Beck-Engelberg (2020), Haslbeck et al. (2020), Lembo (2019) and Vilanova (2020). In their book Narrating is Resistance, Haslbeck et al. (2020) report on their study of 23 Catholic women who were victims of sexual and spiritual abuse in German speaking countries. A psychological study by Makamatine Lembo (2019) uses a qualitative approach to explore the abuse and exploitation of 9 nuns from sub-Saharan Africa. In a recent book, French journalist Constance Vilanova (2020) uses testimonies from nuns in Argentina, India, Italy, France and Sub-Saharan countries to highlight inaction and silence regarding nun abuse. Josephine Beck-Engelberg (2020) documents the results of a survey of women religious, congregations, institutes and organisations that asked questions about abuse undertaken by the German mission society Missio Aache. Survey results revealed serious concerns related to inaction around the abuse of women religious in Asia and Africa.

*2.3. Silencing and Cover-Up*

Despite the seriousness of reported accounts and research findings, the stories of the abuse of women religious have continually been silenced (Allen and Schaeffer 2001; Figueroa and Tombs 2022; Marcel 2007; Sands 2003; Wagner 2018). It was only after extensive media coverage that the reports by O'Donohue and McDonald were even commented on by the Vatican (Figueroa and Tombs 2022; Poggioli 2019). The Chibnall et al. (1998) survey results were not published by the Vatican, by the religious orders that commissioned the study, or any non-Catholic mainstream media channels (Smith 2003). In 2019 Pope Francis was forced to address the issue after complaints were made via female Catholic journalists in Rome (Giuffrida 2019). He admitted that the French community of St Jean had been closed by Pope Benedict in 2005 following allegations of clerical sexual abuse and slavery and forced abortions (BBC 2019) and that there had been multiple claims of the sexual abuse of nuns by clerics from Italy, Africa, Chile, India and France (BBC 2019). In 2018 the Voices of Faith activist group began supporting women religious to tell their stories in a series of public forms, including during the 16 Days of Activism global initiative to end violence against women in 2021 (Voices of Faith n.d.).

There are a number of reasons that could account for the continued cover-up and silencing of accounts and disclosures of the sexual abuse of nuns including media bias, gender norms, and lack of access to external, independent, legal advocacy. These factors intersect to leave women religious reluctant to report and vulnerable to internal church processes and manipulation by clerical and other Church authorities (Marcel 2007). For instance, as noted above, in a recent, highly publicised criminal trial, a bishop in Kerala, India was charged and then acquitted of raping a nun. During the trial, evidence was given that the victim/survivor only reported to police after complaining repeatedly to Church authorities. In handing down his verdict the judge cited that infighting and rivalries within the nun's religious order contributed to the acquittal (Ellis-Petersen 2022). In a Voices of Faith seminar in Rome in 2018, former nun Doris Wagner recalls that when she told her superior that her priest confessor had 'tried to hold and kiss her', the superior tried to excuse the behaviour by saying 'she knew he had a certain kind of weakness for women' (Voices of Faith 2018b, p. 19). As evidence from public inquiries into children who are sexually abused in Church settings has shown, self-blame and shame can discourage the disclosure of sexual abuse (McPhillips et al. 2022). It may also be the case that women religious feel significant shame around sexual abuse incidents and bear responsibility for what occurs.

Durà-Vilà et al. (2013) found that even though all the victim/survivors in their study acknowledged the priest perpetrators were to blame for their abuse, they also acknowledged feeling 'tormented by feelings of shame and guilt' (Durà-Vilà et al. 2013, p. 28), especially when disbelieved by superiors. In some cases, the women religious reported feelings of frustration, anger, mistrust and powerlessness as they were compelled to continue receiving the sacraments and spiritual direction from perpetrators because they either did not disclose the abuse or were not believed (Durà-Vilà et al. 2013). Yet, Durà-Vilà et al. (2013) note that despite blaming and criticising priest perpetrators, the victim/survivors readily excused their congregational leaders. What has not been explored is the status of women religious as consecrated members of the Catholic Church where they may view themselves as a part of the Church leadership, leading to a conflict between needing to protect the reputation of the Church or disclose the harm they experience.

There is a tendency within Church leadership to underplay female perpetrated CSA and interpret adult to adult abuse as consensual or self-induced even when power imbalances exist (Hidalgo 2007). Abuses of power between women religious are often ignored or not understood in a sexual way (Angulo 2021). Brock explains:

> . . . there is a power differential between . . . a religious superior and members of their order. I worked with a young sister once who only ever referred to her religious superior as "my lover". Theirs was a sexual as well as a superior-to-sister relationship. (Brock 2019, p. 37)

Chibnall et al. (1998) found that shared circumstances, including the shared violation of a vow of celibacy and/or chastity permitted women religious who had been sexually abused to temporarily harbour self-delusions as to the motive of the perpetrator and presume mutuality and equal power relations.

The gravity and scandal of the sexual exploitation and harassment of women religious is minimised by the way that power is managed in religious institutes and the Catholic Church more broadly. Internal institutional loyalties normalise abusive behaviours leading to the re-telling of abuse often being framed as criticism of the Catholic Church rather than the reporting of a crime (Angulo 2021). The absence of immediate reporting is often wrongly assumed to be a form of consent (Ellis-Petersen 2022). Without the resources to secure independent legal counsel, women religious are frequently at the mercy of the systems of authority with their religious order for financial and legal support (Marcel 2007). The lack of access to external, independent, legal advocacy can leave women unable to fully argue their cases in public court as demonstrated in the Kerala court case. Thus, making them vulnerable to internal church processes and manipulation by clerical and other Church authority (Marcel 2007).

*2.4. The Media: Part of the Problem*

Media coverage of women's stories of abuse in the Roman Catholic Church has skewed public and Church perceptions of the sexual abuse crisis. Marcel, commenting on the Boston Globe reportage of the sexual abuse crisis (1990–2002), noted that in the Catholic Church, the sexual abuse crisis has been portrayed by sections of the media and the Vatican as a universal crisis that was essentially a problem with homosexual priests, 'thereby (a) conflating a gay (male) sexual orientation with child sexual abusing and (b) implying that the victims all were male' (Marcel 2013, p. 289). Equating the crisis with homosexuality tends to erase female victims, including women religious, from media reporting, making it more difficult for women to disclose clerical sexual abuse (Marcel 2013; Sands 2003; Tuttle 2020). This erasure and obscuration has been exacerbated by generalised representations of rape and female victims in Catholic and secular media (Marcel 2013). Marcel (2013) argues that in the media, men perpetrating sexual violence against women is often framed against naturalised myths of male domination and control and female inferiority and weakness skewing perceptions of abuse. Stereotypes, including a "good girl", "bad girl" dichotomy in crime reporting 'generally include references to the victim's substance abuse, the way she dresses, and her past sexual behaviour—issues used to justify her perpetrator's violence against her based on her own "bad behaviour"' (Marcel 2013, p. 293). This reinforces the false notion that women victims are responsible for male sexual behaviour.

Much like the sexual abuse of women in general across the globe, the media has been largely disinterested in the sexual abuse of women religious (Marcel 2007, 2013; Tuttle 2020). Disclosing their abuse has been particularly traumatic as women were blamed, disbelieved and rejected by their orders. Many nuns were forced to have abortions and perpetrators were protected (Bengali 2018; McPhillips 2019; Colwell and Johnson 2020; Poggioli 2019). A nun might be expected to represent the ultimate "good girl" stereotype and avoid this type of victim blaming, yet even women religious are subject to blame and represented as seductresses. For instance, Rocío Figueroa Alvear, a theologian and survivor of clerical abuse, recounts that when she reported her abuse to deter the beatification of her perpetrator, she was blamed and discredited as a rebellious seductress (Voices of Faith 2018b). The media erasure of women as victims of clerical abuse has meant that despite widespread and systematic abuse of women religious, many victims understood their experiences were rare and isolated. For instance, Wagner asserts 'When I left the community in 2011, I still thought, I was the only nun that had ever been raped by a priest, because I had never heard of similar instances before' (Voices of Faith 2018b, p. 19). Without adequate media attention of the sexual abuse of nuns, the issue has flown under the radar which has allowed Church leaders to shelve the recommendations of research reports and minimise impacts.

*2.5. Public Campaigns: The Role of Protest and Social Media*

Recent public campaigns including the work of Voices of Faith and the #MeToo movement have resulted in a renewed interest and visibility in the sexual abuse of Catholic women and shone a spotlight on the sexual abuse of nuns across the globe. In 2018, Voices of Faith an international Catholic women's organization convened an international meeting in Rome where a number of Catholic nuns, including Wagner and Figueroa Alvear disclosed their sexual assault by clerics (Figueroa and Tombs 2022; Poggioli 2019; Voices of Faith 2018a, 2018b). The event functioned to reveal the power of global protest in breaking the silence around the reality of nun abuse.

The webpage of Voices of Faith cites their vision of 'a prophetic Catholic Church where women's voices count, participate and lead on equal footing with men' (Voices of Faith n.d.). They seek to facilitate innovative events, networking and media outreach to empower women and bring about real change in the patriarchal Catholic Church. Their 2018 event garnered international attention via social media, and the #NunsToo movement was initiated in 2019 (Bengali 2018; McPhillips 2019; Colwell and Johnson 2020; Poggioli 2019).

Social media, activism and media reporting are beginning to break the silence on the abuse of women religious. Advocates for victims/survivors of CSA perpetrated by women religious, however, note that although credible claims of abuse have been made, there has been little media coverage of those cases (Araujo-Hawkins 2021a, 2021b).

## 3. Nuns as Perpetrators of Abuse

Within the Catholic Church the large percentage of perpetrators of CSA have been identified as male clergy members (Benshoff 2019; Hidalgo 2007; Royal Commission into Institutional Responses to Child Sexual Abuse (RCIRCSA) 2017d). As noted above, the focus on male perpetrators and boys as victims has meant that girl victims and female perpetrators have often been overlooked. Over the last 20 years public inquiries investigating CSA in religious institutions and particularly the Catholic Church, have provided a growing body of evidence that women religious have also been perpetrators of physical, emotional and sexual abuse of children in the Catholic Church (Böhm et al. 2014; The Inquiry into Historical Institutional Abuse 1922 to 1995 and The Executive Office (HIAI) 2017; Royal Commission into Institutional Responses to Child Sexual Abuse (RCIRCSA) 2017c, 2017d; Scottish Child Abuse Inquiry (SCAI) 2018, 2019; Te Rōpū Tautoko 2022). Despite the evidence from public inquiries however, there is often incredulity when it comes to believing that women religious could perpetrate sexual abuse. Catholic discourse and gender stereotypes construct nuns as the paradigm of a "good" and "respectful" woman (Marcel 2013), morally superior to the women and children in their care (Gott 2021). Similarly, clergy are perceived as sacred, more "like God" and less likely to sin than other Catholics (McPhillips et al. 2022; Figueroa and Tombs 2022). In a similar way to male clergy perpetrators who are framed as an irregularity, or as 'a few liberal, "gay", American priests' (Marcel 2013, p. 289), women religious who are perpetrators of CSA are also often believed to be an anomaly (Donovan 2002; Hidalgo 2007). The perception of women religious perpetrators as an enigma can contribute to ongoing psychological harm. Indeed, victims/survivors of nun abuse frequently report feelings of isolation and shame, often understanding that they are the only ones to experience harm from a nun perpetrator (Hidalgo 2007; Hill 1995).

The Ryan report into CSA in Ireland documented hundreds of cases of abuse of children by nuns across Ireland in the 20th century (H. McDonald 2009). Much of the abuse occurred in industrial schools, laundries and orphanages between 1930 and 1990 when they were finally closed. The Sisters of Mercy were particularly singled out for their treatment of young women, in particular physical assaults, and humiliation with a smaller proportion of sexual offences (H. McDonald 2009). The Scottish Child Abuse Inquiry (Scottish Child Abuse Inquiry (SCAI) 2018) reported on the sexual abuse of children by Daughters of Charity of St Vincent de Paul nuns at two institutions between 1917 and 1981 and at various institutions run by the Sisters of Nazareth between 1933 and 1984 (Scottish Child Abuse Inquiry (SCAI) 2019). In Australia, the Royal Commission into Institutional Responses to

Child Sexual Abuse (RCIRCSA, 2012–2017) found women religious were perpetrators of CSA (Royal Commission into Institutional Responses to Child Sexual Abuse (RCIRCSA) 2017c).

### 3.1. Incredulity Regarding Female Perpetrators

Even though males make up the majority of perpetrators of CSA in institutional contexts, female perpetrated sexual abuse does occur (Blakemore et al. 2017; Hunt 2006). Very little research, however, examines female perpetrators of CSA in organizational contexts (Blakemore et al. 2017; Darling et al. 2018; Hunt 2006; Spröber et al. 2014). Studies that do exist reveal that between 5 and 31 per cent of all female perpetrated CSA happens in organisational settings such as religious institutions (Hunt 2006). This type of abuse is typically perpetrated by women who are "professional perpetrators"; that is 'those who sexually abuse children in the context of their professional positions' (Darling et al. 2018, p. 5) for example, teachers, nuns and care workers. Furthermore, it is likely that female-perpetrated abuse is significantly under-reported (Blakemore et al. 2017).

The general public tend to be resistant to the conception that women religious are perpetrators of CSA (Hidalgo 2007). Bias and gender norm's function such that it is often difficult for people to conceive of a woman as capable of sex crimes against children (Darling et al. 2018; Kaylor et al. 2021; Reid 2012). Indeed, female perpetrated sex crimes are often denied, minimised or reframed to fit prevailing gender norms and stereotypes that cast females as weaker, more caring and sexually passive compared to males (Reid 2012; Kaylor et al. 2021). Furthermore, vows of chastity and obedience mean that women religious are often constructed as submissive, obedient and self-sacrificing women (Brock 2010). It is commonly thought that women religious do not possess sexual drives and needs and are thus not capable of committing violent and sexual offenses (Hill 1995). As in male perpetration, under-reporting and stereotyping can lead to women religious who commit CSA being viewed as a rogue individual rather than as part of a larger systemic problem of poor management of large-scale abuse by Catholic leaders (Royal Commission into Institutional Responses to Child Sexual Abuse (RCIRCSA) 2017a).

### 3.2. Revelations of Abuse by Victims of Women Religious Perpetrators

Prevalence rates for CSA perpetrated by women religious are difficult to estimate. Hidalgo asserts:

Catholic priests and nuns may be more likely to sexually abuse minors than professionals in similar positions of trust or men and women in the general population. (Hidalgo 2007, p. 43)

Blakemore and colleagues add:

While female perpetrators have also been identified as sexually abusing children in religious institutions, scant discussion of their offending characteristics or victims are found in the published studies. (Blakemore et al. 2017, p. 40)

Given the gendered nature of child violence in the Catholic Church (McPhillips et al. 2022), it is likely that female perpetrated abuse is underreported, and true numbers of women religious perpetrators are unknown.

Evidence from public inquiries indicate that the practices of violence perpetrated by women religious are significant, yet different from those of male clerics. Female perpetrators tend to offend more prominently with physical and emotional forms of violence and punishments are often excessive and gratuitously cruel (The Inquiry into Historical Institutional Abuse 1922 to 1995 and The Executive Office (HIAI) 2017; Royal Commission into Institutional Responses to Child Sexual Abuse (RCIRCSA) 2016, 2017c; Scottish Child Abuse Inquiry (SCAI) 2018, 2019). For instance, in the Report of Case Study 26 regarding the St Joseph's Orphanage, Neerkol which was run by Sisters of Mercy there was extensive evidence of women religious inflicting severe forms of corporal punishment upon children (Royal Commission into Institutional Responses to Child Sexual Abuse (RCIRCSA) 2016). Victims/survivors recalled the nuns as being 'savage and brutal' (Royal Commission into

Institutional Responses to Child Sexual Abuse (RCIRCSA) 2017c, p. 471). Children who had wet their beds were purportedly made to carry their wet sheets to the dining room and stand with them draped over their heads, before being beaten with a cane (Royal Commission into Institutional Responses to Child Sexual Abuse (RCIRCSA) 2017c). One Aboriginal woman described her experience:

> I was very brown, you know, black, and I was scrubbed with a scrubbing brush . . . [a named sister] saw that I was still dirty, and she got the scrubbing brush, and I still have scars on my back today from that because it was really hard, and it was like somebody cutting me with a knife. (Royal Commission into Institutional Responses to Child Sexual Abuse (RCIRCSA) 2017c, p. 318)

The Scottish Child Abuse Inquiry (SCAI) Case Study No. 1 found that children placed in the care of the religious order Daughters of Charity of St Vincent de Paul in Scotland were physically and emotionally abused by women religious (Scottish Child Abuse Inquiry (SCAI) 2018). For some children in the care of sisters being physically hit was a normal aspect of daily life (Scottish Child Abuse Inquiry (SCAI) 2018). Victims/survivors were found to have been physically punished with implements including:

> leather straps, the "Lochgelly Tawse,"[4] hairbrushes, sticks, footwear, rosary beads, wooden crucifixes and a dog's lead. Some also spoke of their mouths having been washed out with carbolic soap as a punishment for using bad language. (Scottish Child Abuse Inquiry (SCAI) 2018, p. 8)

Children who were bed-wetters were physically beaten and humiliated with name calling (Scottish Child Abuse Inquiry (SCAI) 2018). Force feeding was commonplace, and children were often forced to bath publicly and share bath water (Scottish Child Abuse Inquiry (SCAI) 2018). Girls were not educated about menstruation and were punished for blood on bedsheets (Scottish Child Abuse Inquiry (SCAI) 2018). One victim/survivor explained, 'no one told me this was your period, it is a normal part of growing up. What I basically heard was that when I had pain, that was the devil's way of punishing me because I was a bad girl' (Scottish Child Abuse Inquiry (SCAI) 2018, p. 42).

*3.3. Religious Women as Sexual Perpetrators*

As noted above, public inquiry evidence provides a central source of evidence of nuns as sexual abusers. The Australian Royal Commission estimated women religious who abused children made up 5 per cent of the Catholic perpetrator population (Royal Commission into Institutional Responses to Child Sexual Abuse (RCIRCSA) 2017c, p. 82). In New Zealand, the Royal Commission of Inquiry into Abuse in Care report found 3 per cent of the Catholic perpetrator population were women religious (Te Rōpū Tautoko 2022). In Germany, a self-reporting hotline set up by the Catholic Church noted 10.5 per cent of reported child sex abuse was allegedly perpetrated by women religious (Böhm et al. 2014, p. 638; Royal Commission into Institutional Responses to Child Sexual Abuse (RCIRCSA) 2017c, p. 202). Public inquires in Northern Ireland and Scotland provide strong evidence of the abuse of children and young women in orphanages and schools managed by female religious orders (The Inquiry into Historical Institutional Abuse 1922 to 1995 and The Executive Office (HIAI) 2017; Scottish Child Abuse Inquiry (SCAI) 2018, 2019).

The type of CSA perpetrated by women religious includes the inappropriate touching of boys and girls although there is some evidence of more severe forms of sexual assault (The Inquiry into Historical Institutional Abuse 1922 to 1995 and The Executive Office (HIAI) 2017; Royal Commission into Institutional Responses to Child Sexual Abuse (RCIRCSA) 2017c; Scottish Child Abuse Inquiry (SCAI) 2018, 2019). For instance, in the Australian Royal Commission case study of the St Joseph's Orphanage in Neerkol, Queensland a victim/survivor gave evidence that boys were told their penis was "the devil" before being hit around the 'legs and penis with sticks or rulers' (Royal Commission into Institutional Responses to Child Sexual Abuse (RCIRCSA) 2017c, p. 471). In an Australian Royal Commission private

session, a victim/survivor recalled being sexually abused by a number of people in a children's home run by the Catholic Church when he was six years old:

> I'd be made to stand in the large room while one of the priests, and certainly some of the nuns, they'd fondle my genitals and just touch me basically all over my body. I'd be made to stand there either by being hit with hand brooms or straps or any device they'd see fit to hit you with, for a while. And at the same time they'd be groping me. (Royal Commission into Institutional Responses to Child Sexual Abuse (RCIRCSA) 2017c, p. 338)

The report of the Historical Institutional Abuse Inquiry (HIAI) in Northern Ireland found that in the Sisters of Nazareth, Belfast children's home Nazareth House, girls were frequently punished by beating their bare buttocks (The Inquiry into Historical Institutional Abuse 1922 to 1995 and The Executive Office (HIAI) 2017). One victim/survivor:

> explained from personal experience how she was told to lie down on the stage, with one nun holding both her legs in the air, whilst another nun hit her repeatedly across the bottom with a stick, a hairbrush or whatever implement they could get their hands on. (The Inquiry into Historical Institutional Abuse 1922 to 1995 and The Executive Office (HIAI) 2017, chp. 8, para. 157)

The SCAI Case Study No. 2 reported on the provision of residential care for children in Scotland by the Sisters of Nazareth between 1933 and 1984 (Scottish Child Abuse Inquiry (SCAI) 2019). It found that children were sexually abused by sisters, priests, staff, volunteers and other people who had access to the children at each of the four homes run by the Sisters of Nazareth (Scottish Child Abuse Inquiry (SCAI) 2019). One perpetrator, Sister Kevin was found to have emotionally, physically and sexually abused children (Scottish Child Abuse Inquiry (SCAI) 2019). The reported abuse included brutal beatings, denigration, genital touching, and vaginal rape with a broom handle (Scottish Child Abuse Inquiry (SCAI) 2019).

*3.4. Women Religious as Facilitators of Male Perpetrated Abuse*

Women religious were not only perpetrators of abuse but frequently delivered children to male clerics facilitating CSA in Catholic organisations particularly in the orphanage system (The Inquiry into Historical Institutional Abuse 1922 to 1995 and The Executive Office (HIAI) 2017; Royal Commission into Institutional Responses to Child Sexual Abuse (RCIRCSA) 2017c; Scottish Child Abuse Inquiry (SCAI) 2018, 2019). For example, in the SCAI Case Study No. 2, Sister Kevin who was found to have sexually abused children also facilitated the sexual assault of a girl by a number of men including priests (Scottish Child Abuse Inquiry (SCAI) 2019). Sister Kevin not only facilitated CSA but was present when the abuse took place (Scottish Child Abuse Inquiry (SCAI) 2019). In the Australian Royal Commission Case Study 28 evidence was given of an incident of sexual assault that occurred at St Joseph's Orphanage in Ballarat, Victoria where the victim/survivor was delivered to a priest perpetrator by a nun (Royal Commission into Institutional Responses to Child Sexual Abuse (RCIRCSA) 2017c). The victim/survivor reported:

> When I woke up, the priest told me to get out and pushed me out the door, where the nun, who had told me to go to the priest, was waiting outside. She was laughing—big joke to her—and told me to get back to work, maybe because I was walking funny and my bum hurt so much. (Royal Commission into Institutional Responses to Child Sexual Abuse (RCIRCSA) 2017c, p. 331)

Public enquires report that a number of adults, including women religious, were aware of CSA but failed to act or report the abuse (The Inquiry into Historical Institutional Abuse 1922 to 1995 and The Executive Office (HIAI) 2017; Royal Commission into Institutional Responses to Child Sexual Abuse (RCIRCSA) 2017b, 2017c, 2017d; Scottish Child Abuse Inquiry (SCAI) 2018, 2019).

Women religious managed the disclosure of abuse using a variety of techniques including, disbelieving and punishing children, making "pretend" reports, and threatening chil-

dren. For example, in an Australian Royal Commission private session a victim/survivor known as "Gena" recounted 'I did disclose to the nuns. Or a nun, who told me that little girls who tell lies go to hell' (Royal Commission into Institutional Responses to Child Sexual Abuse (RCIRCSA) 2017c, p. 517). Another woman, "Kallie", reported that when she told a nun about her sexual abuse by a priest 'the nun kicked and smacked her for lying and locked her in a cupboard for three days' (Royal Commission into Institutional Responses to Child Sexual Abuse (RCIRCSA) 2017c, p. 546). After the priest sexually abused her again the nun dragged her to the bathroom and scrubbed her genitals with a toothbrush until they bled (Royal Commission into Institutional Responses to Child Sexual Abuse (RCIRCSA) 2017c). Other children testified that when they told the nuns about the sexual abuse, they were punished. One victim/survivor known as "Merle" recalled being strapped for her 'dirty thoughts' and 'talking filth' (Royal Commission into Institutional Responses to Child Sexual Abuse (RCIRCSA) 2017b, p. 146), another related having his head shaved as punishment (Royal Commission into Institutional Responses to Child Sexual Abuse (RCIRCSA) 2017c). The report of the HIAI found evidence of sexual abuse perpetrated by adults and older children in the two homes run by the Congregation of the Poor Sisters of Nazareth (The Inquiry into Historical Institutional Abuse 1922 to 1995 and The Executive Office (HIAI) 2017). It found that there were several instances reported by witness where individual sisters had received reports of CSA or come across improper sexual contact between older boys and younger children (The Inquiry into Historical Institutional Abuse 1922 to 1995 and The Executive Office (HIAI) 2017). Only rarely were instances acted on, with children most often verbally or physically reprimanded for making up such an allegation (The Inquiry into Historical Institutional Abuse 1922 to 1995 and The Executive Office (HIAI) 2017).

*3.5. Media Reporting of Abuse Perpetrated by Women Religious*

As various inquiries and reports are made public and the scale of clerical perpetrated child sexual abuse (CPCSA) emerges, victims/survivors of CSA perpetrated by women religious are beginning to share their stories in the media (Araujo-Hawkins 2021a, 2021b; Tokasz 2018). Media reports and public inquiries, together with social media activist movements such as #MeToo and #ChurchToo have given victims/survivors a vocabulary with which to share their stories. Although globally women religious outnumber priests by more than 200,000, the focus of the media and the Catholic Church when it comes to CSA has been priests and religious men (Araujo-Hawkins 2021b; Dent 2018). Due to the large numbers of male perpetrators, clerical authority, masculinity and priestly celibacy have come under significant scrutiny in public inquiries and scholarly research (John Jay College of Criminal Justice 2011; Sipe 1994). In our review of the literature, we argue that, despite a greater awareness and a growing forum for victims/survivors of female-perpetrated abuse in the Catholic Church to share their stories, the culture and status of female religious life as a contributing factor remains under analyzed. Further research on the topic of women religious who abuse children and adults is needed.

**4. Female Religious Life**

Although there are multiple sites through which women's subjectivity is constructed, the identities of women in the Catholic Church have been produced and shaped via a prevailing patriarchal clericalist system (Schüssler Fiorenza 2016). Herein theological discourses, constructed by male clergy govern the role of women in Catholicism, circulate and are used to maintain patriarchal subject positions and mindsets. These discourses maintain a gendered dualistic model of humanity and church (Schüssler Fiorenza 2016).

The idealized, stereotypical archetype of the "good Catholic woman" as nonsexual, virginal and submissive, creates power asymmetries that support both the subordination of women and the dominance of male clerical power (McEwan 2022). The patriarchal cultural system, which privileges males over females has been used to frame, regulate and police institutional structures and practices including female religious life. As Brock says:

> Subject to male institutional authority, nuns are represented by the Church as living lives of self-sacrifice, devoting themselves wholeheartedly and single-mindedly to God and to the Church's work. (Brock 2010, p. 273)

Religious life is a way of living and being, established via by a formal, solemn and public commitment to God, a religious community/congregation and the Catholic Church (Schneiders 2001). Profession to religious life however is not simply a matter of joining an organisation, rather in becoming a religious woman, individuals give themselves fully to Jesus via a permanent and distinctive state of life.

### 4.1. A Sub-Culture of the Church

Many female religious communities/congregations have existed from the beginnings of Church history. Such sub-cultures, whether they are monastic or worldly in character, or founded by women, are regulated by male-defined Canon Law and the order or institute's own, specific laws and norms which must be approved and ratified by male, clerical authorities (Schneiders 2001). In the Tridentine period[5] of the Catholic Church, female religious communities/congregations were total institutions and the profession of vows which marked an entrance to religious life involved severing almost all relationships not intrinsically related to their religious order or institute (Schneiders 2001). Since the late 1960s, in the post-Vatican II era, many congregations have become more open and integrated into secular society (Schneiders 2001).

Female religious life, however, remains a sub-culture of the Catholic Church, in that it has its own language customs, practices, formation, dress, rules and controls (Brock 2010). Women religious learn about the culture and norms of their community through a spiritual formation process, which itself is governed by Canon law and the particular edicts of the congregation (Schneiders 2001). Historically, formation was a lengthy process of socialisation which involved breaking down previous patterns of behaviour, relationships and expectations as a woman passed from postulate, through the novitiate to profession. In some cases, religious formation was accomplished via a disorientating combination of expectation, praise, restriction, shaming and humiliation (Schneiders 2001). In religious life, as in secular society, an individual is socialised into specific patterns of behaviour, including what is appropriate and what is not, through their lived experience (Brock 2019; McGuire 2008). In some religious communities/congregations, lived experiences of what is acceptable, and what is not, is completely framed by formation processes and the vows of obedience, chastity and poverty.

### 4.2. The Profession of Vows

Upon entering a religious order or institute, each woman professes vows of obedience, chastity and poverty, which are intended to assist a woman to live a life consecrated by God (Johnson et al. 2014; Catholic Church 2000, para. 915).[6] In her study of aging and Catholic nuns, Corwin observes each nun:

> . . . took three vows: a vow of poverty, a vow of chastity, and a vow of obedience. The vow of poverty was designed to teach detachment to material things; the vow of chastity was designed to teach detachment from other humans; and the vow of obedience was meant to teach detachment from self-determination. (Corwin 2021, p. 133)

The vows, which are a sacred commitment approved and sanctioned by the Church, are open to an enormous variety of interpretation among religious orders (Schneiders 2001). Professing vows, however, typically entails a surrender of independence in life and ministry, a situation which can create complex, unequal and gendered power dynamics between women religious and male clerics. In a media interview about the D'Lima et al. (2021) study of women religious in India, Noella de Souza observes:

> The patriarchal and hierarchical system . . . keeps women in a low second place. Many women's congregations . . . comply submissively to what the bishops

desire . . . Seminarians see sisters as menial labor and get used to ordering them around, parish priests see women religious there only to carry out their orders. The negativity on the part of the priest is expressed in actions and attitudes such as refusing to administer the sacraments to the sisters concerned. (cited in Kavi 2021, n.p.)

Indeed, Demasure (2021, p. 284) asserts that 'the patriarchal and clerical system plays a major role in the abuse of both women religious and children', with sacramental blackmail, threats of expulsion from religious congregations and misinterpretation of vows of obedience specific to the abuse of women religious.

The theological goal of the vow of obedience has be understood as a way for women religious, through total obedience to superiors, 'to experience God as the ultimate authority' (Corwin 2021, p. 138). A vow of obedience, however, can compel members to obey congregational leaders without question, to the detriment of personal liberty (Cucci 2020; D'Lima et al. 2021; Figueroa and Tombs 2020, 2022). The recently published guidelines on the challenges of consecrated life the Vatican Congregation for Institutes of Consecrated Life and Societies of Apostolic Life (CICLSAL) confirms:

In certain cases, infantile subjection and scrupulous dependence are promoted instead of collaboration "with an active and responsible obedience". This can betray the dignity of a person to the point of humiliation. (Congregation for Institutes of Consecrated Life and Societies of Apostolic Life (CICLSAL) 2017, para. 25)

In some religious orders, leaders and male clerics, as church authorities, are expected to behave and exercise influence in a "Christ-like" way (Figueroa and Tombs 2022). In orders where superiors and clergy are understood this way there is an increased risk of authoritarianism and a lack of accountability which can be a precursor to abuse (D'Lima et al. 2021; Figueroa and Tombs 2022). Figueroa and Tombs (2022) found an over emphasis of the importance of living in obedience to church authorities was a key factor that enabled the abuse of women religious.

Unequal power dynamics, which can make women religious vulnerable to abuse also place women and children at risk when practices and instructions are accepted from superiors without question. For instance, the SCAI found that in Nazareth Houses in Scotland the inability of the Sisters of Nazareth to question the wisdom of prevailing instructions or practices could be attributed to vows of obedience (Scottish Child Abuse Inquiry (SCAI) 2019). The inquiry found that children suffered abuse and harm because Sisters failed to examine 'matters from their own perspectives or from those of the children' (Scottish Child Abuse Inquiry (SCAI) 2019, p. xi).

The vow of poverty obliges women religious to live with minimal material possessions with the theological intention of bringing them closer to God. In some religious orders, however, women are required to live in harsh and austere conditions. Such conditions, along with religious ideals that placed emphasis on conformity and abandonment of selfhood and glorified suffering contributed to abuse (Clough 2017, pp. 35–36). In this way, the vow of poverty can become deeply embedded in policing the everyday morality of women religious (Clough 2017; Corwin 2021).

Rigidly defined understandings of sex and sexuality work together with the vows of obedience and poverty to establish a gendered, moral code which places boundaries around respectability and sexual activity for women (McPhillips et al. 2022). The profession of a vow of chastity is a commitment to sexual abstinence and prohibits any close or particular attachment to another person. O'Donohue (1994) suggests that the commitment to sexual abstinence is one of the reasons women religious were targeted for sexual activity by male clerics as they were understood as "safe" from HIV/AIDS.

Theologically, a vow of chastity is an expression of lived choice and is symbolic of the self-gifting of one's life to Jesus Christ (Corwin 2021; Schneiders 2001). Corwin notes:

The convent ban on "particular friendships" was part of a lifelong exercise meant to encourage spiritual discipline. This encouraged a shift away from attention to worldly relationships in favor of, in one nun's words, the "one true relationship; the relationship with the divine". (Corwin 2021, p. 135)

The vow of chastity is a fundamental part of female religious life. Trzebiatowska states that the sexual identity of nuns has always been the subject of curiosity and is generally understood in relatively limited terms, 'as a denial of one's sexual identity, or simply as abstinence from sexual intercourse' (Trzebiatowska 2013, p. 208). In Catholicism, however, a vow of chastity sits alongside and is deeply immersed in a belief system which prohibits sexual acts outside a narrowly defined procreative act, performed within a heterosexual marriage (Catholic Church 2000, para. 2351). Under the papacy of John Paul II, ecclesiology was formulated to understand women religious as living out forms of "consecrated femininity" that was in direct complementarity to priestly masculinity (John 1995). The values of consecrated femininity were premised on women embodying the humility, service and self-sacrifice of Mary the mother of Jesus. John Paul II's idealized construction of femininity is reinforced by a theology of complementarianism which positions women in a secondary spiritual relationship with God (McEwan 2022). Clough (2017, p. 127) contends that, in this system, a "Madonna-whore polarity" functions to render women religious "Madonna-like", chaste virgin "Brides of Christ", disconnected from lay women. The presumption is that on entering religious life women will become "sex or gender less" (Trzebiatowska 2013) and lack sexual drives and needs (Clough 2017; Hill 1995). This adds to the earlier described incredulity that women religious can be either victims/survivors or perpetrators of sexual abuse.

In patriarchal systems the coercive quality of sex and gender stereotyping is implicit. The valorisation and internalisation of certain depictions of femininity and womanhood can mean that misogyny is implemented by women, towards women as a form of moral or redemptive correction (Manne 2018; McEwan 2022). For instance, in the Irish Magdalen laundries which were run by various religious orders there were strong links between the abuse of women and girls by nuns as part of a productive policing of morality (Gott 2021). Women and girls were sent to the laundries because they had been or were assumed to be sexually active outside marriage which was considered a deeply immoral state that required redemptive correction. They lived in harsh, ascetic conditions and were shamed, humiliated and put to work by nuns as penance or punishment for their lack of morality (Clough 2017; Gott 2021). By contrast women religious were understood as having distinctive characteristics that made them morally superior and thus more valuable. Yet, in a sexual economy of clericalism, they too were subject to physical and sexual abuse primarily by male clergy. These coercive, punitive mechanisms functioned by way of patriarchal norms and the socio-religious expectations they governed (Manne 2018).

### 4.3. Disciplinary Rituals: Placing Women and Children at Risk

Until recently, some religious orders and institutes used particular forms of disciplinary ritual to enforce norms and laws related to the congregation (Gervais and Watson 2014).[7] Gervais and Watson identify the following forms of discipline in their study of convent life: 'discipline as corporal regulation (e.g., structure, rules, ritual, penance); discipline as punishment (e.g., public apology); and discipline as directive (e.g., vows, vocations)' (Gervais and Watson 2014, p. 279). They argue that resistance and compliance to disciplinary practices co-exist and report various levels of feelings of comfort or conflict within the disciplinary confines of convent living and religious life (Gervais and Watson 2014). Some of the women in their study condemned the rigidity of religious life, while others indicated the advantages of structure and self-restraint (Gervais and Watson 2014).

In orders where disciplinary practices are part of the culture of religious life, children and other vulnerable people in the care of women religious can be at risk of harm. For example, the SCAI Case Study No. 2 reported that children at Nazareth Houses in Scotland were subjected to humiliation and were physically abused until the 1970s using methods

laid out the Directory and Book of Customs published by the Order in 1921 (Scottish Child Abuse Inquiry (SCAI) 2019). The current UK Superior General of the Sisters of Nazareth who ran the homes explained that the order was:

> ... a very strict order and the sisters existed under a very strict regime themselves. ... it looks as if what they lived under, they kind of transferred into their care of the children. (cited in Scottish Child Abuse Inquiry (SCAI) 2019, p. 4)

While this analysis provides insight into the cultures of female religious communities, it does not presuppose that individual agency is absent or lacking but it does suggest that the punishing conditions in which women religious often lived in, and their role in modern Catholic institutional life contributed to both their status as victims and their role as perpetrators.

## 5. The Sexual Economies of Clericalism

In Catholic theology and institutional culture, discursive truths and disciplinary practices function to produce the regulation of the subjectivity of a "woman religious", where she is understood as a particular kind of woman and normatively produced as a docile subject of male power. In this discourse, women religious sacrifice autonomy and independence, accepting a structured life lived in community, and at the service of others. In terms of sexual abuse, this positions women religious as vulnerable and accessible to male desire as part of the sexual economy of clericalism.

The sexual economy of clericalism can be described as a system of both symbolic and real exchange based on unequal power relations between nuns and children, nuns and lay people, nuns and priests and between each other, which produces harmful forms of behaviour when the principles of clericalism are dominant. It is a sexual economy because it is primarily employed in managing the circulation of power between genders with the goal of maintaining the authority, legitimacy and superiority of the priesthood above all other forms of subjectivity. It is maintained through the imposition and practice of canon law and heterodoxic discourse that washes through the institutions, cultural practices and histories of Catholicism. In this section we explore this sexual economy and suggest that it provides insight into the complex situation of women religious historically and currently and may be particularly productive in theorising women religious as both victims and perpetrators of gendered violence.

### Power and Subjectivity

Michel Foucault's (1980, 2003) understanding of power as a force that informs, pervades and challenges conceptions of individual and collective subjectivity is helpful in theorising women religious' subjectivity and agency in Catholicism. In his various genealogies, Foucault (1977, 1988, 1990a, 1990b, 2021) theorises how discourses function within a nexus of power. He upholds that power is not simply an oppressive, top-down influence, rather 'it traverses and produces things, it induces pleasure, forms knowledge, produces discourse' (Foucault 1980, p. 119). This implies that within an institution such as the Catholic Church, subjectivity is socially constructed within a heteronormative discourse of clericalism and is linked to the multiple ways in which power is operationalised in everyday life. Accordingly, Foucault notes:

> Power is employed and exercised through a net-like organization. And not only do individuals circulate between its threads; they are always in the position of simultaneously undergoing and exercising this power. (Foucault 1980, p. 98)

This implies that women religious exercise agency in both productive and oppressive mediums (Butler 2018; Jordan 2000; Taylor 2010).

Positionality within a discursive system of power, however, can also be recognised its own complicity (Foucault 1980). Even what might be understood as practices that liberate or are counter to seemingly dominant power structures can contribute to potential hegemonies. For instance, in The Handmaids Tale (Atwood 2010), the narrator is a handmaid named

"Offred" who, with the other handmaids in Gilead, is controlled and coerced by sexual and psychological violence. Yet, they are not without power. Compelled to sexually service the ruling male elite, as well as police each other, they are part of a complex sexual economy where women are both enforced and enforce, women's bodies are commodified for reproductive purposes but the ruling male elite always retain dominance. In a similar way, the sexual abuse of nuns positions them as vulnerable and accessible to male desire. Nuns are both handmaids of the Church servicing the clerical elite as well as contributing to the regulation of the behaviour of lay women and children.

Power functions because it produces knowledge and shapes self-perceptions, cognitions and preferences so that an individual cannot see or imagine alternatives (Abraham 2019; Ahmed 2010; Foucault 1980). It is notable that nun perpetrators and nuns who are victim/survivors of sexual abuse are often subject to very similar conditions, although relative to each other there were significant power imbalances and different degrees of privilege. Women and girls who were sent to the notorious Magdalene laundries were stripped of their clothes, given alternative names, and were forced to wear a drab uniform (Clough 2017, p. 36). Children in Nazareth Houses operated by the Sisters of Nazareth in Scotland were routinely outfitted in second-hand clothes and shoes and were discouraged from forming relationships with each other (Scottish Child Abuse Inquiry (SCAI) 2019). Similarly, prior to the Vatican II women entering convents were stripped of their identity to enable a complete break with the secular world. They had their heads shaved, exchanged their own clothing for a habit and were given new, often male, religious names (Clough 2017, p. 36).

Women's participation in religious life can be understood as support for patriarchal traditions while maintaining the marginalisation of women in the Church. However, in a study of Cairo's urban women's mosque movement, Saba Mahmood (2012) proposes that agency is not simple resistance or opposition to social norms but modalities of action. For some women participation in religious life and compliance with patriarchal norms, which might be understood as passive or docile compliance, is a form of agency. For instance, for some contemporary women religious, a vow of chastity is a deliberate choice to willingly forgo marriage and sex within a hedonistic post-modern society and culture. Religious life can be viewed as the only viable alternative to marriage. In some cases, it provided a safe haven from sexual abuse by family or clerical men (Hidalgo 2007).

This kind of life-choice does not solely belong to women religious but is produced and contingent on the discursive framework of Catholicism. Holloway's (1984) notion of "investment" is helpful here. Holloway (1984) critiques Foucault's assumption that power equally subjugates men and women, and instead proposes that in milieus where there are concurrent but opposing discourses on masculinity and femininity, the perception of fulfillment and/or incentive influences support of a particular discourse. For example, in Catholicism, women religious who devote their life to the hegemonic construction or ideal of the "nun", can benefit by being recognised as the ultimate "good Catholic woman". Women adopting the standards of patriarchal systems, however:

> . . . does not signal necessarily a reversal of the hierarchy or power base; it is more likely a reflection of women's participation in, upholding of and negotiation within the patriarchal status quo. (Abraham 2019, p. 79)

Leimgruber (2022) suggests that women who work in pastoral employment are at greater risk of sexual abuse and that the risks of violence are poorly understood in caring professions that particularly rely on theology to guide relationships. McEwan (2022) found the more women are engaged in Church life, the more likely they are to be victims of spiritual abuse. Hence, women religious can continue to experience pain, alienation and anger as they negotiate their lived experiences within the constraints of the institutional Church (Gervais 2012b). For instance, the reality of gender-based restrictions for women religious in ministry were recently reinforced when the Vatican criminalised the ordination of women and announced it as a "grave crime" in Church law, on a par with the sexual

abuse of minors (Gervais 2012a; McElwee 2021). In a study of Canadian women religious, sisters recounted:

> instances of sexist discrimination, humiliating disrespect in public, patriarchal interference, and patronising denials by bishops and priests of their work as school principals, teachers, pastoral associates, feminist authors, liturgical coordinators, and conference organizers. (Gervais 2012a, p. 8)

Figueroa and Tombs (2022) report on three main factors that enable sexual abuse against nuns: clericalism, sexism and authoritarianism. Yet, individuals can exercise agency and re-position themselves in relation to dominant expressions of power (Butler 2018; Jordan 2000; Taylor 2010).

Many religious women have resisted patriarchal, clericalist forces. Their subjectivity positions them outside of normative forms of femininity which has provided freedoms as well as restrictions allowing them to be at the forefront of movements of social change including challenging Church authority and power (Corcoran 2017; Gervais 2012a; Voices of Faith 2018a, 2018b). The work of women religious has been instrumental in building networks of social capital and welfare and providing labour for the infrastructure projects of the Catholic Church particularly in the colonial diaspora as noted in the example of Mary McKillop's order. Many women religious challenge Church precepts and strategically create alternative spiritual and liturgical initiatives to bypass male dominated institutional constraints (Corcoran 2017; Gervais 2012a, 2012b). Their contribution to the global social welfare (health, education, migration, etc.) of citizens is significant.

## 6. Conclusions

We have argued that the sexual economies of clericalism are explicitly linked to the patriarchal, sexist and misogynistic ideologies, theologies and social practices that are widely accepted and normalised in the Catholic Church. Such social systems and institutions allocate a subordinate place to women and girls in relation to men and distribute rights and obligations unequally, often visiting harsh or adverse social consequences on women as a way of enforcing and policing gendered social norms (Manne 2018). The sexual commodification of women religious provides one answer as to how women can be both entrapped and command power in a limited oppressive clerical economy.

As a social category, nuns remain highly unorthodox in modern times, transgressing the sexual norms of traditional femininity and sacrificing the "benefits" of traditional heterodoxic femininity; they are subject to clerical authority and abuse, as well as having access to limited power over other women and children. Their status is contested and paradoxical.

Shifting this discourse into a sexual economy of clericalism allows us to see such paradoxes within a network of power where the labour and bodies of nuns have been commodified and utilized to help build a global Catholic empire, uphold a male clerical elite, and where their sexual agency has been repressed through the currencies of theologies premised on sexual shame and guilt.

Within this economy nuns can be both victims of patriarchal power and perpetrators of abuse and the work ahead involves lifting these stories out of the silence in which they have been embedded for decades. Clearly, much more work needs to be done to understand the matrix of abuse and perpetration amongst women religious and could include ethnographies and histories of religious orders which prioritize and make visible the impacts and uses of sexual economies of clericalism, and large-scale surveys aimed at understanding the global prevalence of abuse and its impacts.

The larger story of nuns in relation to the sexual abuse crisis and child violence is still to be unearthed and analysed.

**Author Contributions:** Author Contributions: Conceptualization, K.M.; Formal analysis, K.M. and T.M.; Funding acquisition, K.M.; Investigation, K.M. and T.M.; Methodology, K.M. and T.M.; Project administration, K.M.; Writing original draft, K.M. and T.M.; Writing—review & editing, K.M. and T.M. All authors have read and agreed to the published version of the manuscript.

**Funding:** This research received no external funding.

**Conflicts of Interest:** The authors declare no conflict of interest.

## Notes

[1]  In Catholicism women enter religious life by publicly professing vows of poverty, chastity and obedience (Catholic Church 2000, para. 915). Women living a religious life who are in the world, actively engaged in apostolic duties are generally referred to as "sisters", with women living in closed, monastic communities generally called "nuns". Nuns and sisters both profess religious vows and live a communal life in a religious order or institute (Gervais and Watson 2014; Tuttle 2020). The term "women religious" is inclusive of both "nuns" and "sisters" and in this article the terms are used interchangeably.

[2]  Australian Royal Commission into Institutional Responses to Child Sexual Abuse (RCIRCSA) (2012–2017); Te Rōpū Te Rōpū Tautoko (2022); Voices of Voices of Faith (2018a, 2018b); Scottish Child Abuse Inquiry (SCAI) (Scottish Child Abuse Inquiry (SCAI) 2018, 2019); The Inquiry into Historical Institutional Abuse 1922 to 1995 and The Executive Office (HIAI) (The Inquiry into Historical Institutional Abuse 1922 to 1995 and The Executive Office (HIAI) 2017); Commission to Inquire into Child Abuse (CICA) (Commission to Inquire into Child Abuse (CICA) 2009a, 2009b).

[3]  Clericalism is the idealization of male clerical power and by extension the idealization of the ecclesiology and theology of the Catholic Church (Royal Commission into Institutional Responses to Child Sexual Abuse (RCIRCSA) 2017b). Clericalism manifests chiefly through the belief and practice that male clerics (deacons, priests and bishops) are set apart from the laity (including women religious) by nature of their ordination as "ontologically changed", superior and closer to God (Plante 2020). As part of a patriarchal system of power, clericalism allows a small group of elite men to have absolute authority and is 'linked to a sense of entitlement, superiority and exclusion, and abuse of power' (Royal Commission into Institutional Responses to Child Sexual Abuse (RCIRCSA) 2017b, p. 43). Clericalism has been linked to the sexual abuse of children (Plante 2020), women and nuns (Demasure 2021) by male clerics.

[4]  '"Tawse" is an old Scots word which means the thongs of a whip; the leather strap, although a single piece at one end, would, at the other end, have been split into two or three tails. The "Lochgelly Tawse" was a particular type of tawse, supplied to schools in Scotland by John J. Dick, Leather Goods, a manufacturing company based in Lochgelly, Fife.' (Scottish Child Abuse Inquiry (SCAI) 2019, p. 14).

[5]  The Tridentine period is the four hundred years leading up to Vatican II (also referred to as the Second Vatican Council 1962–1965).

[6]  Some religious communities/congregations take additional vows (Johnson et al. 2014; Catholic Church 2000, para. 915).

[7]  Disciplinary rituals were common in female religious orders and institutes until the mid to late 1960s (Gervais and Watson 2014). Anecdotal accounts and abuse narratives indicate that certain disciplinary rituals and practices continued until the 1980s.

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
