# Peer review of "The Sexual Economies of Clericalism: Women Religious and Gendered Violence in the Catholic Church"

_religions, doi:10.3390/rel13100916_

Round 1

Reviewer 1 Report

I think it is an excellent article. There are few studies about religious women as victims of abuse but almost nothing has been done regarding the intersection of religious women as victims and perpetrators. The article gives an exhaustive review of the studies done so far regarding nuns and sexual violence. It provides a brilliant theoretical framework and sociological  analysis to understand the situation of women religious as victims and perpetrators.  On the other hand, the historical background, the cover-up and the influence of the media are fundamental elements that the article brings giving a very comprehensive picture of the topic.   Within the text there are just two minor errors: the word AUTHOR is present twice so it is necessary to eliminate it or write the reference.  

Author Response

Author will be amended to correct names once the peer review process is completed.

Reviewer 2 Report

Review (doc) See also PDF File 

Abstract

Line 18: the article does not consider the situation in the Vatican

Line 20 : It is not clear to me what the methodological framework consists of. Please explain in the text

Text

Line 72 : O’Donue 1994 s not a study but a Memo. Ir. could be called "report". See also Personal Memo from Sr. Maura O’Donohue MMM:Meeting at SCR, Rome, 18 February 1995

Line 88: Leimgruber (not Leimbruger) also to be corrected in the bibliography

Line 91: If the original article cannot be found: at least the name of Lucetta Scarafia should be mentioned

Line 95 : The article Figueroa and Tombs refer to (The guardian) date from February 5 2019. So the meeting cannot have taken place in March. see Nicole Winfield AP, February 5, 2019  : Pope publicly acknowledges clergy sexual abuse of nuns

https://apnews.com/article/pope-francis-ap-top-news-international-news-crime-sexual-abuse-397f2c76afc04532908035a66ccaacc8

Line 250 : Sands is not mentioned in the bibliography

Line 354: systemic problem, needs some explication

Lines 358-360; “more likely to sexually abuse than professionals in similar positions” The source is outdated (2007) Please check relevant sources that are up to date

Line 491-492 : one cannot state that research remains scarce based on a source in 1995 (Hill) 27 years have passed since 1995

Line 552 : women religious and other clerics. Women religious are NOT clerics. This has huge consequences also for the elaboration of the topic of your paper. “The sexual economies of Clericalism.” It cannot include religious sisters as clerics. So, they can be considered as victims of clericalism, meaning victims of deacons, priests, bishops (which includes cardinals) but they cannot be considered perpetrators under the umbrella of “clericalism” 

Clerics are : any individual who has been ordained, including deacons, priests, and bishops. A more inclusive term could be : religious

Line 610-611: References to a book such as Clough (2017) do not mention the page, which makes it quite difficult to check . e.g. line 611 : states they need to BECOME "sex out genderless" ..and therefore they do not POSSESS sexual drives and needs. I wanted to read the original text, since becoming and possessing seemed to be to be a contradiction. Maybe better reframe

613 : It is not because you control the sexual drives or that you are a-sexual that you cannot become a victim of abuse.

Line 656-674: clericalism needs to be defined. The abuse by nuns cannot be part of “clericalism” only the abuse by deacons/priests/bishops of children and adults. A solution could be that the author writes that the power nuns have nuns over children and that religious superiors have power over other sisters has some similarities of the power priests have. However in the hierarchy of the Catholic Church the power of priests which includes sacramental power is much more important. Nuns cannot administer sacraments which puts them under the priests in relation to power. 

Please also explain clearer what exactly the sexual economies are.

Line 671: define political economy

Conclusion, please see my notes in the text

The original sources need to be consulted : Figueroa and Tombs are cited 25 times! the literature on the topic is rather limited, so original works (still) can be consulted. Referring to Lembo, you also refer to Figueroa and Tombs. If the original source has been consulted, I wonder why also referring to F and T. which is a secondary source. The way the reference to Lembo is indicated givres the impression her work has been consulted. 

The fact that the referrals to book often do not include pages, makes it quite difficult to check : e.g. Corwin, Clough.

Following sources have not been consulted and are not part of your study and might be interesting to be part of your study

Demasure, K., Vulnerability of Women Religious in the Context of Abuse,  in Studia Canonica, 55/1-2, (2021) 275-288. 

Dura-Vila, Gloria et al. “The Dark Night of the Soul: Causes and Resolution of Emotional Distress among Contemplative Nuns.” Transcultural Psychiatry 47 (2010): 548 – 570. 

Flynn, K. A., The Sexual Abuse of Women by Members of the Clergy, McFarland, 2010. 

Haslbeck, B., Heyder, R., Leimgruber, U., Sandherr-Klemp, D., Erzählen als Widerstand. Berichte über spirituellen und sexuellen Missbrauch an erwachsenen Frauen in der katholischen Kirche, Aschendorff Verlag GmbH & Co KG, Münster, 2020.

Reisinger, D., #NunsToo. Sexueller Missbrauch an Ordensfrauen - Fakten und Fragen, in: Stimmen der Zeit 236 (6/2018), S. 374−384

Author Response

Please see attached document for response to Reviewer #2.

Round 2

Reviewer 2 Report

Line 189 Beck-Engelberg (2020)  is available in English and is not waiting translation

line 478- 481 lay out to be corrected (indentation)

Line 568 Johnson font needs to be corrected